# Interrupted Time Series Analysis: Patient Characteristics and Rates of Opioid-Use-Disorder-Related Emergency Department Visits in the Los Angeles County Public Hospital System during COVID-19

**DOI:** 10.3390/healthcare11070979

**Published:** 2023-03-29

**Authors:** Emily Johnson, Sarah Axeen, Aidan Vosooghi, Chun Nok Lam, Ricky Bluthenthal, Todd Schneberk

**Affiliations:** 1Department of Emergency Medicine, Keck School of Medicine, University of Southern California, Los Angeles, CA 90033, USA; 2Keck School of Medicine, University of Southern California, Los Angeles, CA 90033, USA; 3Department of Population and Public Health Sciences, Keck School of Medicine, University of Southern California, Los Angeles, CA 90032, USA

**Keywords:** COVID-19, opioid use disorder, social determinants of health, buprenorphine, emergency medicine

## Abstract

The COVID-19 pandemic led to disruptions in care for vulnerable patients, in particular patients with opioid use disorder (OUD). We aimed to examine OUD-related ED visits before and during the COVID-19 pandemic and determine if patient characteristics for OUD-related ED visits changed in the context of the pandemic. We examined all visits to the three public safety net hospital EDs in Los Angeles County from April 2019 to February 2021. We performed interrupted time series analyses examining OUD-related ED visits from Period 1, April 2019 to February 2020, compared with Period 2, April 2020 to February 2021, by race/ethnicity and payor group. We considered OUD-related ED visits as those which included any of the following: discharge diagnosis related to OUD, patients administered buprenorphine or naloxone while in the ED, and visits where a patient was prescribed buprenorphine or naloxone on discharge. There were 5919 OUD-related ED visits in the sample. OUD-related visits increased by 4.43 (2.82–6.03) per 1000 encounters from the pre-COVID period (9.47 per 1000 in February 2020) to the COVID period (13.90 per 1000 in April 2020). This represented an increase of 0.41/1000 by white patients, 0.92/1000 by black patients, and 1.83/1000 by Hispanic patients. We found increases in OUD-related ED visits among patients with Medicaid managed care of 2.23/1000 and in LA County safety net patients by 3.95/1000 ED visits. OUD-related ED visits increased during the first year of the COVID pandemic. These increases were significant among black, white, and Hispanic patients, patients with Medicaid managed care, and LA County Safety net patients. These data suggest public emergency departments served as a stopgap for patients suffering from OUD in Los Angeles County during the pandemic and can be utilized to guide preventative interventions in vulnerable populations.

## 1. Introduction

The COVID-19 pandemic led to dramatic disruptions in care which exacerbated the challenge of access for patients facing structural vulnerability, in particular people with opioid use disorder (OUD) [1,2]. Closures of outpatient addiction clinics, cessation of harm reduction services, and lack of access to support groups are potential contributors to worsening outcomes for patients with OUD during the pandemic; alarmingly, rates of opioid overdose appear to be increasing [3,4,5,6,7,8,9,10]. In Los Angeles (LA), a county disproportionately affected by high rates of COVID-19, chronic housing insecurity, and substance use, the Department of Public Health reported a 48% increase in accidental drug overdose deaths during the first five months of the pandemic [11]. These deaths were largely driven by fentanyl-related overdose, with rates of 117% compared with the same period in 2019 [11].

Pandemic-related shutdowns were associated with an initially sharp reduction in emergency department (ED) visits. Nationwide, ED visits dropped over 40% after March 2020 [12,13,14]. At the same time, the profound disruption in usual sources of care for patients with and without insurance left EDs serving as a critical access point for some vulnerable groups [4]. Los Angeles safety-net hospitals noted a decrease of 37% in ED visits, with higher ED utilization among older, black, and male patients [15]. While some patients may have had access to established care providers through telemedicine or other means (such as electronic health messaging), these forums were less likely to reach the socially vulnerable—patients without phones, internet, housing, English language proficiency, or established sources of care [16,17,18,19,20,21]. As cities across the United States begin to examine the effects of limiting healthcare access during the pandemic on patients with OUD, few have examined the impact on safety net emergency departments by patients who found themselves without other access to care.

The emerging literature on the intersectional factors comprising structural vulnerability highlights the multifactorial nature of health disparities in analogous populations [22]. This lens of structural vulnerability provides a deeper illustration of the unique adversities for patients disproportionately impacted by both the OUD and COVID-19 epidemics [8,15,23,24]. The patients served by the LA County safety net EDs face multiple structural vulnerabilities—they are largely uninsured or publicly insured, experience poverty, and are from communities of color, and many lack stable housing. Since the Patient Protection and Affordable Care Act, many patients have been enrolled in Medicaid managed care plans, where access is often limited by gatekeeper primary care providers and limited specialty networks [25,26]. The LA County Emergency Departments are housed within three large, public, academic referral hospitals located in the North, South, and East Los Angeles areas, and collectively serve more than 310,000 patients per year [27]. Given the disproportionate burden of morbidity and mortality from both COVID and OUD on vulnerable populations, examining the use of these safety net EDs offers insight into the unique interaction between COVID and OUD in a population who may be highly susceptible to disruptions in already-fragile access to healthcare.

### Timeline

LA County began to report increased cases of COVID-19 in February 2020 [28]. In March of 2020, a state of emergency was declared in LA and stay-at-home orders were issued. Restrictions were lifted in May 2020, but by June, daily cases began to climb, forcing another closure in July 2020. Though restaurants and businesses closed, cases began to climb in November, with the largest peaks in cases in January 2021 [28].

We hypothesized, based on the closure of outpatient facilities and our clinical experience during the closures, that there would be increased reliance on the ED for patients with OUD compared to pre-COVID levels, and that both black and Hispanic patients compared with white patients would have higher increases in ED utilization for OUD-related care during the COVID period [15]. We examined the visit-related rates and patient characteristics of patients presenting to LA County safety net Emergency Departments for OUD-related encounters to better understand the populations impacted by the intersection of the COVID and opioid crises.

## 2. Materials and Methods

This study analyzes OUD-related ED utilization at the three safety net hospitals in LA County from April 2019 through February 2021. We examined all ED encounters and defined OUD-related ED visits as those with any of the following: (1) an ICD-10 discharge diagnosis related to OUD, (2) visits where patients were administered buprenorphine or naloxone while in the ED, and (3) visits where a prescription for buprenorphine or (4) naloxone was given on discharge (details on ICD-10 codes are available in Appendix Table A5). A report from the electronic health record allowed us to capture these data across all sites, including medications administered in the ED and prescriptions associated with the ED visit upon discharge. We augmented our administrative dataset with additional data fields both reported by patients or assessed by nurses during triage. Acuity level is assessed by the triage nurse using the emergency severity index (ESI), a nationally recognized five-level algorithm where 1 = highest acuity (for example, a pulseless patient) and 5 = lowest acuity (for example, a routine prescription refill request) [29]. Additional clinical fields include mode of arrival, categorized as arrival by ambulance vs. other (public transit, private auto, walked, etc.). Fields imported from the administrative dataset include age, gender, race/ethnicity, primary payor, and housing status (experiencing homelessness vs. housed).

Similar to emergency departments across the country, LAC + USC Medical Center in LA County observed a precipitous drop in ED utilization after safer-at-home orders in March 2020 [15]. Given the numeric reduction in the number of patients presenting to the ED, our key outcome was the rate of OUD-related visits per 1000 ED encounters. To understand how ED utilization for OUD varied with the onset of COVID-19, we estimated differences in patient populations by COVID period. To determine not only how levels of utilization changed, but also trajectories of utilization, we employed an interrupted time series analysis to examine pre-COVID rates of ED visits for OUD compared with OUD-related visits during COVID. We defined the pre-COVID period as extending from April 2019 to February 2020 (hereafter, Period 1); during COVID was defined as April 2020 through February 2021 (hereafter, Period 2). We excluded the month of March 2020 as it covers a period both before and after COVID was recognized and treated as a widespread threat in LA County. We present interrupted time series (ITS) estimates showing the rate of OUD-related encounters per 1000 ED visits and ITS estimates for specific subgroups. As a result, for example, the rates of visits by race/ethnicity groups or housing status are additive.

We also examined ITS analyses by group of payor, as defined into three groups: non-LA County safety net, Medicaid managed care, and LA County safety net population, which includes patients of particular interest to the LA County public healthcare system based on their primary payor (LA County safety net empaneled, undocumented primary care plan, uninsured, and Hospital presumptive eligibility). The Medicaid managed care group includes only those in Full Scope Medicaid Managed Care plans. The non-LA County safety net population includes private insurance, VA/Tricare, other empaneled Medicaid managed care, and Medicare. We performed all statistical tests in Stata version 15 with α set to 0.05. The study was approved by the USC Institutional Review Board.

## 3. Results

### 3.1. Population Characteristics of the Sample

Table 1 illustrates the summary statistics for the sample, comparing Period 1 (pre-COVID, April 2019–February 2020) with Period 2 (during COVID, April 2020–February 2021). There were 520,991 ED visits during the two-year observation period; 5919 ED visits (1.14%) were related to OUD. While there was a reduction in overall ED visits from 296,642 in Period 1 to 224,349 in Period 2, we saw a slight increase in the number of OUD-related ED visits, from 2763 in Period 1 to 3156 in Period 2.

Over that same period, the total population utilizing the safety net hospitals of LA County was 65% Hispanic/Latino, 13% black, 34% in Medicaid managed care, 7.2% privately insured, 8.2% uninsured, and 8.1% experiencing homelessness. A higher proportion of patients with OUD-related encounters compared with non-OUD-related encounters were male (64.6% vs. 51.9%), age 19–39 (37.6% vs. 31.4%), 40–64 (48.1% vs. 43.3%), black (14.7% vs. 12.9%), white (11.9% vs. 4.6%), in a Medicaid managed care plan (43.8% vs. 33.4%), and experiencing homelessness (30.2% vs. 7.9%). Associated clinical characteristics of OUD-related ED visits included higher proportions brought in by ambulance (41.8% vs. 16.5%) and kept as inpatient (48.0% vs. 14.5%) (See Appendix Table A4 for additional descriptors of the sample).

### 3.2. Interrupted Time Series Analysis

Figure 1 displays the results of our interrupted time series analysis for all patients and Figure 2 by race/ethnicity. We include the interrupted time series (ITS) analysis results in Table 1. In Period 1, 9.47 (9.12, 9.82)/1000 ED encounters were related to OUD. In Period 2, this increased by 4.43 (2.82, 6.03) per 1000 encounters.

### 3.3. Race/Ethnicity

In Period 1 vs. Period 2, there was no significant increase in the proportion of OUD-related encounters among white patients (12.3% and 11.6%, respectively), a difference of −0.64% (−2.30%, 1.01%). There was a statistically significant increase in the proportion of OUD-related ED encounters among black patients, from 13.0% in Period 1 to 16.1% in Period 2, a difference of 3.1% (1.29%, 4.90%). We saw no significant change in the proportion of OUD-related ED visits among Hispanic/Latino patients, from 44.5% in Period 1 to 41.1% in Period 2, a difference of −3.35% (−5.88%, 0.83%).

In our interrupted time series (ITS) analysis, there was, however, a statistically significant level increase in OUD-related ED visits among black, white, and Hispanic groups, with the greatest level change among Hispanics, an increase of 1.83 (0.49, 3.16) per 1000, followed by an increase among black patients of 0.92 (0.44, 1.40) OUD-related ED visits per 1000, and a smaller level increase among white patients 0.41 (0.10, 0.72) per 1000.

### 3.4. Visit Acuity

As shown in Table 1, we found no statistically significant difference in the proportion of high-acuity patient encounters for OUD-related ED visits, which was 37.2% in Period 1 and 35.9% in Period 2, a difference of −1.34 [−3.80, 1.12]. However, we did note a statistically significant increase in low-acuity OUD visits, from 22.0% in Period 1 to 27.5% in Period 2, a difference between Period 2 and Period 1 of 5.50% (3.30%, 7.70%). Similarly, in our ITS analysis (Table 2), we noted a significant positive change in the trend in low acuity visits of 0.29 (0.06–0.51).

### 3.5. Payor Group

We noted no significant change in the proportion of OUD-related ED visits among the LA County Safety net population, from 81.2% Period 1 to 80.7% in Period 2 (difference of −0.48 [−2.48, 1.5]. Nor did we find any difference among the proportion of OUD-related ED visits for Medicaid managed care patients, from 42.8% in Period 1 to 44.7% in Period 2, a difference of 1.92% (−0.61, 4.45).

However, in our ITS analysis, among the LA County Safety net population, we saw a level increase of 3.95 (2.29–5.60) ED visits per 1000 encounters in Period 2 compared with Period 1 (see Table 3). Similarly, we saw in the Medicaid managed care population a level increase of 2.23 (1.34, 3.12) OUD-related ED visits per 1000.

## 4. Discussion

This study demonstrates that OUD-related ED visits increased disproportionately and dramatically as a proportion of all ED visits after COVID-related shutdowns in LA County. The unadjusted rate of OUD-related ED visits increased during COVID by about 50% (from 9 to 14 per 1000 ED visits).

### 4.1. A Closer Look at Vulnerable Groups

A closer look at the impact of COVID on low-acuity ED visits for OUD-related care (diagnosis codes such as medication refills for buprenorphine or wound care) reveal not only an underlying lack of access to outpatient addiction care, but may also illustrate the disproportionate impact of pandemic-related clinic closures among vulnerable populations with OUD. In these data, we see that many patients with OUD presented to ED waiting rooms during the pandemic in need of routine care (see Appendix Table A1, Table A2 and Table A3 for further breakdown of low acuity visits by race/ethnicity) [30]. We also note that there was the greatest level increases in our ITS analysis among black and Hispanic/Latino patients. This closer look at the characteristics of patients presenting for OUD-related care to the ED before and during COVID allows us to visualize the specific structural factors of patients with OUD in LA that make them uniquely vulnerable to times of limited healthcare access and societal strain [8,31].

The widespread closure of clinics and doctors’ offices in the wake of statewide stay-at-home orders likely contributed to our finding of higher predicted probabilities of OUD-related ED visits for patients across insurance groups. Specifically, the level of OUD-related ED visits for patients both in the safety net and with Medicaid managed care increased significantly, reflecting the critical role of the emergency department for those with limited care networks in times of societal strain.

These findings illustrate the structural vulnerability of this primarily minoritized and low-income population with OUD who are reliant on safety net EDs for care. It is essential, furthermore, to note that health outcomes for patients with OUD are highest among persons of color, who also experienced the highest overall mortality during the pandemic [24,32]. This ED utilization study highlights the pivotal role that EDs play in providing care to patients with limited healthcare access and demonstrates how ED utilization may serve as an indicator of healthcare access gaps even for patients with insurance. It also represents an opportunity for health systems to examine opportunities for outreach to vulnerable populations whose access is most threatened when systems of care are disrupted during a pandemic or other crises.

### 4.2. Limitations

LA County public hospitals are not as dependent on commercial billing as other EDs, which may lead to less stringent documentation requirements compared with other health systems. Subsequently, visit-related diagnoses for admitted patients may be limited to their most critical complaint, such as altered mental status or respiratory failure, which could also lead to missed cases. This would likely underestimate the true number of OUD-related encounters; however, we do not expect differential documentation trends between Period 1 and 2. We attempted to compensate for limited diagnosis coding with data on medication delivery and ED prescriptions to capture OUD-related patient encounters without diagnoses codes. Some demographic categories are based on patient-reported values. We also note there was a change in the method of recording housing status during Period 1, so we include our ITS analysis for patients experiencing homelessness in the appendix (Table A1, Table A2 and Table A3, Figure A1 and Figure A2) with this caveat. Where possible, we used administratively verified measures (e.g., acuity and insurance payor) rather than self-reported measures. Less than 1% of data for the variables age, race/ethnicity, and insurance status were missing. We centered the data around the most dramatic phases of the COVID pandemic in 2020; thus, the data presented span only until 2021, and may not reflect current emergency department utilization. Finally, while this unique view into the LA County public hospital safety net system provides a valuable perspective into a primarily low-income, marginalized urban population, it has a low proportion of patients with private insurance and may not generalize to rural communities or other populations.

## 5. Conclusions

This analysis of the three large safety net hospital emergency departments in a county severely impacted by high rates of COVID-19 highlights the critical role emergency departments played for patients with OUD. As has been evidenced throughout the COVID-19 pandemic, minoritized and vulnerable populations were disproportionately impacted. In the face of future social crises and pandemics, attention must be paid to ways to maintain access to preventative OUD care for patients, especially those who are structurally vulnerable. Additionally, health systems across the country should continue to equip emergency departments to embrace their role as a critical access point for medication-assisted treatment and harm reduction services. For those disproportionately impacted by both OUD and COVID, such as those served in our hospitals, these data are the canary that should call attention to the fragility of the current safety-net for the structurally vulnerable.

## Figures and Tables

**Figure 1 healthcare-11-00979-f001:**
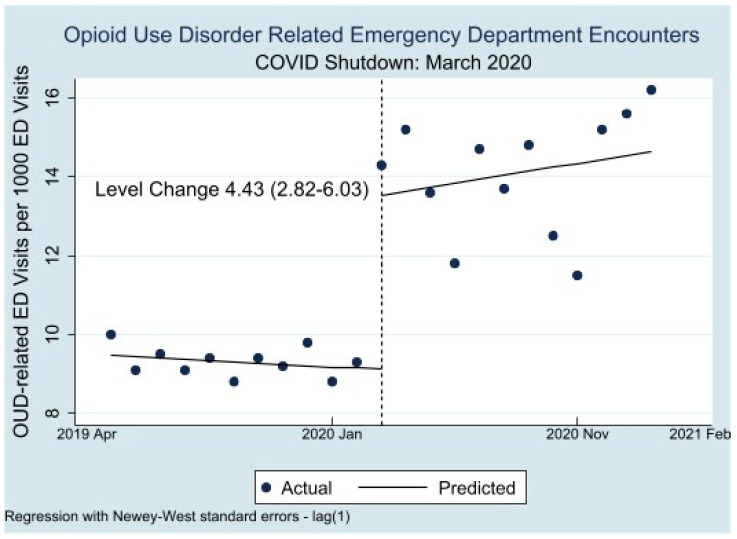
Interrupted Time Series of OUD-related LA County Emergency Department Visits by Month.

**Figure 2 healthcare-11-00979-f002:**
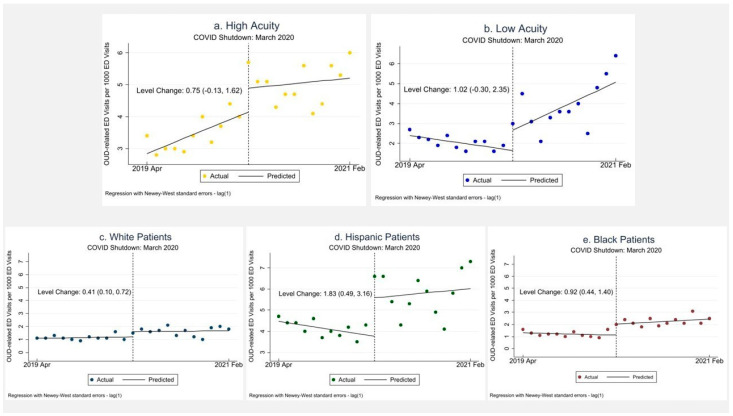
Interrupted Time Series for Opioid-Use-Disorder-related Emergency Department visits (**a**) High-Acuity Visits; (**b**) Low-Acuity Visits; (**c**) White Patients; (**d**) Hispanic Patients; (**e**) Black Patients.

**Table 1 healthcare-11-00979-t001:** Summary Statistics: Demographic and Clinical Characteristics of Opioid-Use-Disorder-Related Emergency Department Patient Encounters Pre- and During COVID-19.

Measure	Period 1:	Period 2:	Difference
	April 2019–February 2020	April 2020–February 2021	Period 2–Period 1
	N (% of All Visits)	N (% of All Visits)	% Difference [95% CI]
All Visits OUD Visits	296,642 (100)	224,349 (100)	-
	2763 (0.93)	3156 (1.41)	**0.48 [0.42, 0.53]**
	N (% of OUD Visits)	N (% of OUD Visits)	% Difference [95% CI]
Race/Ethnicity			
Black	360 (13.0)	509 (16.1)	3.10 [1.29, 4.90]
White	339 (12.3)	367 (11.6)	−0.64 [−2.30, 1.01]
Hispanic/Latino	1229 (44.5)	1298 (41.1)	−3.35 [−5.88, 0.83]
OUD-Related Acuity			
High	1028 (37.2)	1132 (35.9)	−1.33 [−3.80, 1.12]
Low	607 (22.0)	867 (27.5)	5.50 [3.30, 7.70]
Payor			
LA County Safety Net	2244 (81.2)	2548 (80.7)	−0.48 [−2.48, 1.52]
Medicaid Managed Care	1183 (42.8)	1412 (44.7)	1.92 [−0.61, 4.45]
Other Characteristics			
Male Gender	1868 (67.6)	1956 (62.0)	−5.63 [−8.07, −3.19]
Experiencing Homelessness	800 (29.0)	989 (31.3)	2.38 [0.04, 4.72]

**Table 2 healthcare-11-00979-t002:** Interrupted Time Series Results by Race/Ethnicity and Patient Acuity.

	All Encounters	White	Black	Hispanic	Acuity: High	Acuity: Low
Total N	5919	732	913	2622	2264	1527
Pretrend	−0.033 (0.027) [−0.088, 0.023]	0.01 (0.015) [−0.021, 0.041]	−0.017 (0.028) [−0.075, 0.040]	**−0.065** (0.023) [−0.113, −0.018]	**0.118** (0.033) [0.048, 0.188]	**−0.068** (0.019) [−0.109, −0.028]
Level change in April 2020	**4.427** (0.766) [2.824, 6.030]	**0.410** (0.148) [0.101, 0.719]	**0.920** (0.229) [0.441, 1.399]	**1.828** (0.638) [0.492, 3.164]	0.747 (0.417) [−0.126, 1.62]	1.021 (0.633) [−0.304, 2.345]
Change in trend (post–pre)	0.133 (0.0143) [−0.166, 0.432]	−0.005 (0.029) [−0.067, 0.056]	0.056 (0.033) [−0.012, 0.124]	0.105 (0.121) [−0.149, 0.359]	−0.090 (0.067) [−0.230, 0.051]	**0.286** (0.108) [0.061, 0.512]
Preperiod level	**9.472** (0.167) [9.123, 9.822]	**1.086** (0.070) [0.940, 1.233]	**1.304** (0.134) [1.024, 1.585]	**4.472** (0.109) [4.244, 4.702]	**2.845** (0.209) [2.409, 3.282]	**2.40** (0.124) [2.135, 2.655]
Postestimation Commands						
Weekly change in outcome during COVID	0.100 (0.140) [−0.192, 0.393]	0.005 (0.024) [−0.046, 0.056]	**0.039** (0.018) [0.001, 0.077]	0.040 (0.115) [−0.200, 0.280]	0.0287 (0.058) [−0.094, 0.151]	0.218 (0.105) [−0.002, 0.439]

Notes: values in **bold** are statistically significantly different from 0 at the *p* < 0.05 level. The total N reflects the observations used in making these estimates; the number of observations in the regressions is 23 for the 12 months preceding March 2020 and the 11 months following March 2020.

**Table 3 healthcare-11-00979-t003:** Interrupted Time Series Results by Payor Group.

	All Encounters	Medicaid Managed Care	LA County Safety Net
Total N	5919	2595	4792
Pretrend	−0.033 (0.027) [−0.088, 0.023]	−0.022 (0.032) [−0.088, 0.045]	−0.032 (0.019) [−0.072, 0.008]
Level change in April 2020	**4.427** (0.766) [2.824, 6.030]	**2.229** (0.427) [1.337, 3.122]	**3.945** (0.792) [2.287, 5.603]
Change in trend (post–pre)	0.133 (0.0143) [−0.166, 0.432]	0.059 (0.086) [−0.121, 0.238]	0.048 (0.145) [−0.254, 0.351]
Preperiod level	**9.472** (0.167) [9.123, 9.822]	**4.1** (0.155) [3.775, 4.425]	**7.723** (0.137) [7.436, 8.009]
Postestimation Commands			
Weekly change in outcome during COVID	0.100 (0.140) [−0.192, 0.393]	0.037 (0.078) [−0.127, 0.200]	0.0164 (0.143) [−0.282, 0.315]

Notes: values in **bold** are statistically significantly different from 0 at the *p* < 0.05 level. The total N reflects the observations used in making these estimates; the number of observations in the regressions is 23 for the 12 months preceding March 2020 and the 11 months following March 2020. Patients may be included in both Medicaid and safety net.

## Data Availability

The data presented in this study was obtained under a data use agreement with the Los Angeles County Department of Health Services and cannot be shared due to sensitive nature. Similar data can be requested from the Los Angeles Department of Health Services.

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
