# Peer review of "Interrupted Time Series Analysis: Patient Characteristics and Rates of Opioid-Use-Disorder-Related Emergency Department Visits in the Los Angeles County Public Hospital System during COVID-19"

_healthcare, 2023, doi:10.3390/healthcare11070979_

Round 1

Reviewer 1 Report

Dear Authors, 

Many thanks for your submission and allowing me to review the paper.

The paper is very good and well written and highlights the issues of treatment access due to both the challenges facing this population group and those affected plus the added challenges encountered due to the pandemic.

The information is well presented, the figures used where a bit small and could be made larger making them clearer.

I have a couple of small comments

Line 111 - you use ITS for the first time but do not explain what this is, can you please put extended here and use ITS thereafter?

Tables are presented as split over pages can this be resolved (and this could be the typesetting but it makes it harder to read)

Table 3 - Under Total N states All encounters 5919 but Medicaid is 2595 and Safety Net 4792 - these totals do not add up but this could be my understanding and it should be clarified if patients can claim both Medicaid and Safety Net.

Line 214 - "Shuttering" would this be better changing to "closure"

Well written and well presented.

Kind regards

Reviewer 2 Report

“Interrupted time series analysis: patient characteristics and rates of opioid use disorder – related Emergency Department visits in the Los Angeles County public hospital system during COVID-19”

This paper aims to examine the impact of the COVID-19 pandemic on OUD-related ED visits in public safety-net hospitals in Los Angeles County. The study used interrupted time series analyses to compare OUD-related ED visits before and during the COVID-19 pandemic and determined if patient characteristics for OUD-related ED visits changed in the context of the pandemic.

I find the subject interesting for the Opioid Crisis during the COVID-19 Pandemic special issue. However, the manuscript does not offer the required depth, cohesion, clarity, and comprehensiveness. A revision is needed to enhance the contribution to an acceptable level. Therefore, I recommend minor revision. Afterward, a resubmission is encouraged if comments are addressed and the draft is re-written meticulously. I sincerely hope that my assessments are helpful for the authors. My main concerns and comments are summarized below:

Main Concerns:

1.      Limited generalizability: The study focuses on three safety-net hospitals in LA County, which may not be representative of other hospitals or regions. This limits the generalizability of the study's findings and makes it difficult to draw broader conclusions about the impact of the COVID-19 pandemic on OUD-related ED utilization.

2.      Limited definition of OUD-related ED visits: The authors define OUD-related ED visits as those with a discharge diagnosis related to OUD, visits where patients were administered buprenorphine or naloxone while in the ED, or visits where a prescription for buprenorphine or naloxone was given on discharge. While this is a reasonable starting point for defining OUD-related visits, it is possible that some patients with OUD may not have been identified using these criteria. For example, patients with OUD who do not receive buprenorphine or naloxone during their ED visit may not be captured by this definition. Authors mentioned that they also consider ICD-10 discharge diagnosis related to OUD, but they should specify which specific ICD-10 discharge diagnosis codes related to OUD they considered in their study. This would provide clarity and transparency for readers who may want to replicate or compare the results with other studies.

3.      ED Utilization: Authors do not consider the possibility that changes in ED utilization may have been influenced by factors other than COVID-19, such as changes in healthcare delivery or patient behavior.

4.      Introduction and Literature review: The literature review section could be better. Some major papers were missed. Please start with the following papers:

·         Haley, Danielle F., and Richard Saitz. "The opioid epidemic during the COVID-19 pandemic." Jama 324.16 (2020): 1615-1617.

·         Rikin, Sharon, et al. "Changes in outpatient opioid prescribing during the COVID-19 pandemic: an interrupted time series analysis." Journal of Primary Care & Community Health 13 (2022): 21501319221076926.

·         Sahebi-Fakhrabad, Amirreza, Amir Hossein Sadeghi, and Robert Handfield. “Evaluating State-Level Prescription Drug Monitoring Program (PDMP) and Pill Mill Effects on Opioid Consumption in Pharmaceutical Supply Chain.” Healthcare. Vol. 11. No. 3. Multidisciplinary Digital Publishing Institute, 2023.

·         Sedney, Cara L., et al. "Assessing the impact of a restrictive opioid prescribing law in West Virginia." Substance Abuse Treatment, Prevention, and Policy 16.1 (2021): 1-12.

·         Fakhrabad, Amirreza Sahebi, et al. "The Impact of Opioid Prescribing Limits on Drug Usage in South Carolina: A Novel Geospatial and Time Series Data Analysis." arXiv preprint arXiv:2301.08878 (2023).

Other Concerns:

1.      It is important for authors to be aware of proper citation style when writing a paper. One key element of citation style is ensuring that references are mentioned before closing the sentence. For example, it is incorrect to write "xxx.[y]" and correct to write "xxx [y]." This small but important detail can help to ensure that references are properly attributed and that readers can easily locate the source material.

2.      Page 1, line 41: citation should not be in superscript format.

Overall, the study's findings are important and suggest that there is a need for preventative interventions and policies to address the impact of the COVID-19 pandemic on vulnerable populations. However, more research is needed to determine if the findings of this study can be generalized to other hospitals or regions, and to understand the specific interventions and policies that may have influenced OUD-related ED visits during the COVID-19 pandemic. So I recommend minor revision, and I encourage the authors to resubmit the paper after addressing the comments.

Author Response

Please see attachment for detailed response. Thank you for your feedback. 
